# Replication of Missense *OTOG* Gene Variants in a Brazilian Patient with Menière’s Disease

**DOI:** 10.3390/genes16060654

**Published:** 2025-05-28

**Authors:** Giselle Bianco-Bortoletto, Geovana Almeida-Carneiro, Helena Fabbri-Scallet, Alberto M. Parra-Perez, Karen de Carvalho Lopes, Tatiana de Almeida Lima Sá Vieira, Fernando Freitas Ganança, Juan Carlos Amor-Dorado, Andres Soto-Varela, Jose A. Lopez-Escamez, Edi Lucia Sartorato

**Affiliations:** 1Meniere Disease Neuroscience Research Program, Faculty of Medicine & Health, School of Medical Sciences, The Kolling Institute, University of Sydney, Sydney, NSW 2064, Australia; gisellebiancobortoletto@gmail.com (G.B.-B.); amparraperez@ugr.es (A.M.P.-P.); 2Laboratory of Human Molecular Genetics, Center for Molecular Biology and Genetic Engineering—CBMEG, State University of Campinas—UNICAMP, Campinas 13083-875, SP, Brazil; hfs@unicamp.br; 3Programa de Pós-Graduação em Ciências Médicas, Faculty of Medical Sciences, State University of Campinas—UNICAMP, Campinas 13083-970, SP, Brazil; 4Department of Biochemistry and Tissue Biology, Laboratory of Paracrine Signalling in Tissue Organisation—SPOT, Programa de Pós-Graduação em Biologia Molecular e Morfofuncional, Biology Institute, State University of Campinas—UNICAMP, Campinas 13083-862, SP, Brazil; g289650@dac.unicamp.br; 5Postdoctoral Researcher Program, Faculty of Medical Sciences, State University of Campinas—UNICAMP, Campinas 13083-887, SP, Brazil; 6Otology & Neurotology Group CTS495, Division of Otolaryngology, Department of Surgery, Instituto de Investigación Biosanitaria, ibs.GRANADA, Granada, Universidad de Granada, 18012 Granada, Spain; 7Sensorineural Pathology Programme, Centro de Investigación Biomédica en Red en Enfermedades Raras, 28029 Madrid, Spain; 8Department of Otolaryngology and Head and Neck Surgery, Federal University of São Paulo, São Paulo 04038-032, SP, Brazil; kclopes04@gmail.com (K.d.C.L.); tatiana.vieira@unifesp.br (T.d.A.L.S.V.); ffgananca@gmail.com (F.F.G.); 9Department of Otolaryngology, Hospital Can Misses, 07800 Ibiza, Spain; juancarlosamordorado@gmail.com; 10Division of Neurotology, Department of Otorhinolaryngology, Complexo Hospitalario Universitario, 15706 Santiago de Compostela, Spain; andres.soto@usc.es; 11Department of Surgery and Medical-Surgical Specialities, Universidade de Santiago de Compostela, 15782 Santiago de Compostela, Spain; 12Health Research Institute of Santiago (IDIS), 15706 Santiago de Compostela, Spain

**Keywords:** Meniere disease, hearing loss, *OTOG* gene, molecular bioinformatics

## Abstract

Ménière’s Disease (MD) is a chronic inner ear disorder defined by recurring episodes of vertigo, fluctuating sensorineural hearing loss, tinnitus, and/or fullness in the ear. Its prevalence varies by region and ethnicity, with scarce epidemiological data in the Brazilian population. Although most MD cases are sporadic, familial MD (FMD) is observed in 5% to 20% of European cases. Through exome sequencing, we have found a rare missense variant in the *OTOG* gene in a Brazilian individual with MD with probable European ancestry (chr11:17599671C>T), which was previously reported in a Spanish cohort. Two additional rare missense heterozygous *OTOG* variants were found in the same proband. Splice Site analysis showed that chr11:17599671C>T may lead to substantial changes generating exonic cis regulatory elements, and protein modelling revealed structural changes in the presence of chr11:17599671C>T, chr11:17576581G>C, and chr11:17594108C>T, predicted to highly destabilize the protein structure. The manuscript aims to replicate genes previously reported in a Spanish cohort, and the main finding is that a Brazilian patient with MD also has variants previously reported in familial MD, supporting *OTOG* as the most frequently mutated gene in MD.

## 1. Introduction

Ménière’s Disease (MD, MIM 156000) is a rare chronic inner ear disorder that manifests through recurrent episodes of vertigo, fluctuating sensorineural hearing loss (SNHL), tinnitus, and aural fullness. Its prevalence varies across regions and ethnic groups worldwide, ranging from 34 to 190 per 100,000 adults, with familial aggregation and higher occurrence among women and Caucasians [1,2]. Epidemiological studies in Western European and East Asian populations have shown significant differences in familial aggregation and associated co-morbidities in MD [3,4,5]. However, the lack of epidemiological data in the Brazilian population makes genetic studies an arduous task.

Although most MD cases are considered sporadic, familial Meniere’s Disease (FMD) has been reported in 5–20% of cases of European descent [6]. The current diagnostic approach relies on clinical criteria outlined by the Barany Society in the 2015 international guidelines, emphasizing clinical stratification and the identification of incomplete phenotypes [1]. Exome and genome sequencing studies combining gene burden tests and segregation analyses support MD as a polygenic condition [7,8,9]. Around 20 genes have been linked to familial MD [6], including SNHL genes demonstrating autosomal dominant, recessive, and digenic inheritance patterns [10,11], supporting the concept of genetic heterogeneity in this condition.

The *OTOG* gene, encoding Otogelin, an extracellular protein in the tectorial membrane in the organ of Corti [12,13], is emerging as a key gene in familial MD [14]. Several missense variants in *OTOG* have been identified in Spanish families with MD, showing aggregation in the coding sequence of constrained regions, which may explain the higher MD prevalence in Europeans [11,14].

This exome sequencing study aims to identify novel candidate genes in Brazilian patients with MD. Our results replicate a rare missense variant chr11:17599671C>T in the *OTOG* gene previously reported in a Spanish cohort in a Brazilian MD patient, supporting the role of *OTOG* in the molecular pathophysiology of MD.

## 2. Materials and Methods

### 2.1. Hearing Profile

Audiograms from the worse ear were obtained from patients and analyzed in R Studio (v.4.4.2) using a regression equation to estimate hearing onset and outcome. The coefficient of determination (*R*^2^) and *p*-values validated the model.

### 2.2. Whole Exome Sequencing (WES)

WES was performed on DNA samples from 23 unrelated Brazilian Menière’s Disease (MD) patients. Libraries were prepared with Nextera Rapid Exome Enrichment Kit^®^ (Illumina, Inc., San Diego, CA, USA) and sequenced (2 × 100 base pairs paired end, ≥100× coverage) on HiSeq™ 2500 (Illumina). Reads were aligned to the GRCh38/hg38 using Burrows–Wheeler Aligner (BWA), and variants were called with DeepVariant and filtered for minor allele frequency (MAF) < 0.001 (Genome Aggregation Database, gnomAD v4.1.0) and CADD score > 20. The Integrative Genomics Viewer (IGV) assessed read quality.

### 2.3. Rare Variant Identification

Rare variants were retrieved from previously published sporadic MD (SMD, *n* = 511) [11] and familial MD (FMD, *n* = 100) [8] datasets (non-Finnish European MAF < 0.001, CADD score > 20). Pathogenicity was assessed per American College of Medical Genetics and Genomics (ACMG) and Association for Molecular Pathology (AMP) guidelines [15] applied for Genetic Hearing Loss [16]. Missense variant density in the *OTOG* coding sequence was determined using a 200 base pair (bp) sliding window template previously reported [14], identifying high-density regions (HDRs) and low-density regions (LDRs), with the latter also called constraint coding regions (CCRs), per gnomAD v2.1.

#### 2.3.1. Splice Site Prediction

Human Splice Finder Pro [17] was used to predict donor/acceptor sites, exonic and intronic splicing enhancers (ESEs/ISEs), exonic and intronic splicing silencers (ESSs/ISSs), and branch points.

#### 2.3.2. Protein Modelling

Canonical UniProt sequences were modelled using Modeller [18] v10.5 through homology-based reconstruction using a template of the WT protein previously reported [14]. Model validation included Discrete Optimized Protein Energy (DOPE), Genetic Algorithm 341 (GA341), and molecular probability density function (molpdf). Additional validation was performed via the Structural Analysis and Verification Server (SAVES v6.1; https://saves.mbi.ucla.edu/) (accessed on 25 October 2024 and 30 October 2024) for stereochemical quality and atomic interactions [19,20]. Structural analysis and images were obtained in PyMOL v3.0 [21].

DynaMut2 [22] and MuPro [23] assessed variant impact on protein stability (ΔΔG). MuPro confidence score (CS) ranged from −1 to 1, with CS < 0 indicating destabilization and CS > 0 indicating stabilization. DynaMut2 classified variants as destabilizing for negative predicted free energy change (ΔΔG_pred) values and stabilizing for positive ΔΔG_pred values.

## 3. Results

### 3.1. Clinical Description

This study focuses on a male proband who meets definitive MD diagnostic criteria outlined by the Barany Society, with a 13-year disease course characterized by recurrent vertigo attacks (25–30 episodes), persistent left-sided tinnitus, fluctuating SNHL, and aural fullness. A comprehensive neurotological evaluation, incorporating video head impulse testing (vHIT) and serial audiometry, was conducted (Table 1), which confirmed bilateral SNHL without vestibular hypofunction. His medical history included well-controlled hypertension and autoimmune disease, with no familial history of auditory or vestibular disorders.

#### Hearing Profile

The hearing profile of the Brazilian proband and three Spanish patients carrying the *OTOG* variant chr11:17599671C>T showed no significant hearing loss progression at any frequency (Appendix A).

### 3.2. Rare OTOG Variant Replicated in Brazilian Sample

The missense variant chr11:17599671C>T (NFE MAF = 0.005) was replicated in both cohorts. Two additional rare *OTOG* missense variants were identified in the Brazilian proband, chr11:17576581G>C (NFE MAF = 0.00001) and chr11:17594108C>T (NFE MAF = 0.0004). The three of them were classified as Likely Pathogenic per ACMG guidelines, and chr11:17599671C>T showed a lower allelic frequency, while chr11:17576581G>C and chr11:17594108C>T showed a higher allelic frequency in the Brazilian population (ABraOM) when compared to non-Finnish (gnomAD_AF) and Spanish (CSVS_AF) populations (Table 2).

Variants chr11:17599671C>T and chr11:17576581G>T were predicted to destabilize the protein in both in silico methods employed, while chr11:17594108C>T showed conflicting results. When analyzed together within Otogelin, mimicking the proband’s protein, DynaMut2 stability prediction was −1.9 kcal mol^−1^, indicating a higher degree of destabilization (Appendix A).

Furthermore, the selected variants are located in biologically constrained low-density regions (LDRs), suggesting low protein tolerance to variation (Figure 1). We were unable to determine whether they are cis or trans-variants, as it was not possible to obtain samples from the patient’s parents, as they are deceased.

Using HSF Pro, analysis of the three *OTOG* variants showed significant exonic cis-element alterations for chr11:17599671C>T, including the disruption of five ESEs and the formation of two ESEs and seven ESSs (Appendix A).

All three missense variants are located in and around the von Willebrand factor type D 3 (VWFD3) domain. The VWFD, EGF-like, and TIL domains are found in the head of the protein, while the CTCK domain is located in the tail (Figure 2a,b).

In silico analysis revealed that both wild-type and mutated amino acids are in loop regions. The variant chr11:17599671C>T (p.P1240L) was predicted not to form any contacts in the WT structure; however, the mutated amino acid formed a hydrophobic contact with the Proline at p.1236 (Figure 2c).

For chr11:17594108 C>T (p.P1129L), WT Proline forms several hydrophobic contacts with p.W1144, while the variant Leucine creates new hydrophobic and polar contacts with p.W1144, p.A1520, and p.N1131 (Figure 2d).

The variant chr11:17576581G>T (p.G850R) replaced Glycine with Arginine at position 850, which is predicted to create three hydrophobic contacts with p.I599 (Figure 2e) and alter polar interactions, leading to a short α-helix at p.841–843 (Appendix A).

## 4. Discussion

This study replicates a missense variant in a Brazilian patient with SMD previously reported in two individuals and one family with MD in Spain.

There is an overload of missense variants in the *OTOG* gene in FMD [24]. The variant chr11:17599671C>T has been associated with moderate-to-severe flat hearing loss (60 dB) from early onset, involving all frequencies with minor low-frequency variations over time. Interestingly, it has not been reported in non-European individuals. The Brazilian proband, a male with no MD family history and unknown European ancestry, was screened for additional *OTOG* variants. Two rare heterozygous missense variants, chr11:17599671C>T and chr11:17594108C>T, were identified, neither previously associated with the disease.

The tectorial membrane (TM), an acellular layer above the organ of Corti, plays a key role in sound transmission by enhancing cochlear feedback [25]. The TM is generated by cochlear supporting cells [26], and it is organized on the apical membrane of the supporting cells facing the lumen of the scala media [27]. Structurally, it consists of four collagen proteins (II, V, IX, XI), that forms the main structural framework and several non-glycoproteins, and proteoglycans, including α-tectorin, β-tectorin, otogelin, otogelin-like and CEACAM16 [28,29]. Otogelin, encoded by *OTOG*, as a key non-collagenous glycoprotein [30]. *OTOG* knockout mice have revealed its importance in auditory and balance functions, showing phenotypes like detachment of otoconial membranes, structural defects in the TM’s fibrillar network, and reduced TM resistance to sound stimuli [30]. Otogelin stabilizes the TM by interacting with its constituent fibres, and *OTOG* mutations can cause moderate nonsyndromic SNHL [28].

When a protein residue mutates, new bonds replace existing ones, significantly altering the protein’s structure if the new residue’s properties differ greatly. These changes affect the conformation and bonding network around the mutation site, depending on the residues and their environment [31].

All variants are located around (chr11:17599671C>T and chr11:17576581G>T) and within (chr11:17594108 C>T) the VWFD 3 domain in Otogelin protein. The TM likely regulates Ca^2+^ levels near hair cell stereocilia and in mechanoelectric transduction channel adaptation, with the VFWD domains of Otogelin binding Ca^2+^ ions as a reservoir [32]. Rare variations in Otogelin may create new electrostatic interactions, altering their 3D structure [14].

Our study has a limitation since we could not obtain DNA samples from the proband’s relatives for segregation analysis, but the finding of three rare variants in the *OTOG* gene in the same proband cannot be explained by chance.

## 5. Conclusions

We have found a patient with MD with three rare missense variants in the *OTOG* gene, which were predicted to destabilize the protein and alter its structure. This is the first time the variant chr11:17599671C>T has been reported in non-Spanish patients with MD. These results support the *OTOG* gene as a key player in the pathophysiology of MD.

## Figures and Tables

**Figure 1 genes-16-00654-f001:**
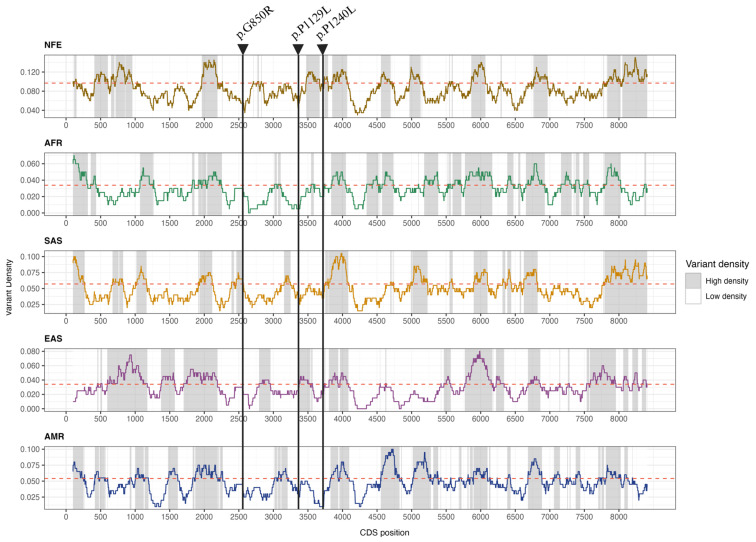
Variant density profile along the *OTOG* CDS in the NFE, AFR, SAS, EAS, and AMR populations calculated with a 201 bp sliding window. The threshold density for each population based on the anticipated number of variants within CDS is indicated by the red dashed line.

**Figure 2 genes-16-00654-f002:**
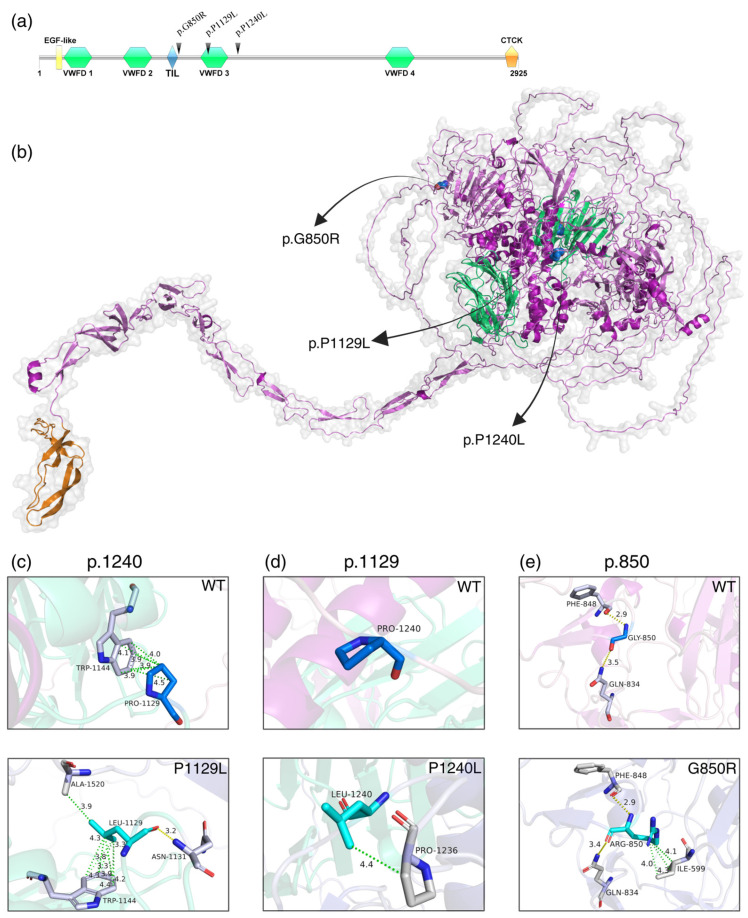
(**a**) Otogelin protein domains: EGF-like (epidermal growth factor), VWFD (von Willebrand factor type D), TIL (trypsin inhibitor-like, cysteine-rich domain) and CTCK (C-terminal cystine knot) (created with IBS v.1.0.3 software). (**b**) Modelled Otogelin protein showing the three selected variants. (**c**–**e**) Structural differences between wild-type (WT, in blue) and mutated (in cyan) Otogelin residues at positions p.850, p.1129, and p.1240. Interactions between amino acids are represented by yellow (polar contacts) and green (hydrophobic) dotted lines, with respective distances measured in Ängstrom. It is possible to observe that all mutated amino acids create new bonds with neighbouring residues (in grey).

**Table 1 genes-16-00654-t001:** Summary of the phenotype observed in the patient with rare variants in *OTOG*.

Age of Onset	Duration of Disease	Comorbidities	Hearing Thresholds	Hearing Stage	Caloric Test	Functional Scale
53 years	13 years	High blood pressure Hypothyroidism Psoriasis	RE * 10 dB LE * 53 dB Fluctuation in LE *	3	11.5% asymmetry (normal)	4

* RE = right ear; LE = left ear.

**Table 2 genes-16-00654-t002:** *OTOG* missense variants found in the proband. Chr11:17599671C>T was replicated in the Spanish cohort.

Variant	chr11:17599671C>T	chr11:17576581G>T	chr11:17594108C>T
ID	rs117005078	rs61734214	rs7936354
Consequence	Missense	Missense	Missense
Amino acid change	p.Pro1240Leu	p.Gly850Arg	p.Pro1129Leu
CADD	33.00	27.6	24.1
gnomAD_AF	0.005990	0.00001918	0.0004865
CSVS_AF	0	0	0
ABraOM_AF	0.003416	0.005978	0.034586
ACMG *	Likely Pathogenic	Likely Pathogenic	Likely Pathogenic
DynaMut2 prediction	Destabilize (−0.49 kcal mol^−1^)	Destabilize (−0.44 kcal mol^−1^)	Destabilize (−0.7 kcal mol^−1^)
MuPro prediction	Decrease stability (−0.24 kcal mol^−1^)	Decrease stability (−0.76 kcal mol^−1^)	Increase stability (0.12 kcal mol^−1^)

* ACMG criteria according to Oza et al. [13].

## Data Availability

The *OTOG* variants chr11:17599671C>T, chr11:17576581G>T and chr11:17594108 C>T have been deposited in ClinVar (SUB15056944, accession number pending).

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
