# Peer review of "Replication of Missense OTOG Gene Variants in a Brazilian Patient with Menière’s Disease"

_genes, 2025, doi:10.3390/genes16060654_

Round 1

Reviewer 1 Report

Comments and Suggestions for Authors

In the manuscript entitled “Replication of missense OTOG gene variants in a Brazilian cohort of Menière’s Disease”, the team of Dr. Lopez Escamez, an international Meniere’s Disease (MD) expert, identified in one Brazilian patient with MD, by exome sequencing, 3 rare missense variants (chr11:17599671C>T, chr11:17576581G>C, and chr11:17594108C>T) in the OTOG gene. The latter encodes Otogelin, an extracellular glycoprotein specific to the acellular membranes of the inner ear implicated in the auditory and balance mechanotransduction processes. One of the 3 rare variants (chr11:17599671C>T) has previously been characterized in a Spanish cohort of familial MD. Analysis of this variant showed significant exonic element alterations. Structural modeling predicts that these 3 variants could alter the stability.of the otogelin protein. Altogether, these findings emphasize the key role of OTOG in  familial MD.

The article is well written and presents convincing results that reinforce the involvement of the OTOG gene in the familial MD pathophysiology. However, I have a few comments:

Line 2: A rewording of the title would be desirable to reflect the data more accurately, as the missense OTOG gene variants were detected in only one patient in the cohort analyzed. The word “cohort” is overstated.

Line 124: Clinical description: It would have been useful to include a summary table of clinical and paraclinical data (notably audiograms, caloric test results - often impaired in  familial MD - and vHIT - frequently normal in familial DM) for the Brazilian patient as well as for the three Spanish carriers of the mutation, especially as in figure S1 hearing in the worse ear remains stable. Did the authors perfom MRI on the Brazilian patient to identify if the mutation is associated with radiological evidence of hydrops? This information would be informative as the patients' audiometric profile, as presented in figure S1, seems relatively stable over time.

In figure S1, it is written that magenta and violet dots represent the two Spanish patients but only the magenta dots are visible; the red dots are not present for all frequencies tested. In addition, the magenta color does not appear in the graphs for the 4000 and 6000 Hz frequencies. Are the data missing or not available? The authors should check these elements and modify the legend accordingly to ensure a clear reading of the data.

Additional minor comments:

Line 125: the authors wrote that the male proband meets definitive MD diagnostic criteria. They should clarify if this diagnostic is based on clinical criteria outlined by the Barany Society.

Line 158-159 (Figure 1): Please specify what the dotted red line refers to.

Lines 173-175 (Figure 2 c-e): Please specify in the legend the meaning of the yellow and green dotted lines and the associated numerical values.

In table S2 please correct the typo” Slpicing” by” splicing”

In figure S2A, panels B and C, it would be advisable for the authors to specify in the legend what are the yellow dotted lines and associated numerical values.

Reviewer 2 Report

Comments and Suggestions for Authors

Bianco-Bortoletto et al., described three missense mutations in OTOG gene in one Brazilian Meniere Disease patients in their manuscript “Replication of missense OTOG gene variants in a Brazilian cohort of Menière’s Disease”. They found that one mutation, chr11:17599671C>T, was previously reported in a Spanish cohort and the other two mutations are only found in this Brazilian proband. They further analyzed the effect of those mutations on mRNA splicing and protein stability, and they concluded that those three mutations additively compromise the stability of Otogelin at the protein level. Genetic factors in Meniere Disease are of great interest for researchers and physicians, and this manuscript confirms the possible involvement of OTOG mutations in pathogenesis of Meniere disease. However, there are a few issues with the current version of the manuscript.

First, the author did not provide any data to show those three mutations were found in the same allele, thus the additive effect of missense mutations on OTOG at the protein level is questionable.

Second, the statement in line 142 regarding the allele frequency of those variants is not accurate. AF for chr11:17599671C>T is lower and for other two mutations are higher in Brazilian population compared to other populations, as shown in table 1.

Third, legends for figures and tables can be improved to help readers. For example, acronyms in figure 1 should be explained in the figure legend.

Reviewer 3 Report

Comments and Suggestions for Authors

This study reports the replication of missense OTOG gene variant in a Brazilian patient with Meniere's disease.  Interesting report

Materials and Methods and results : Study focuses on only one proband. Therefore the title of the manuscript needs to be revised.
Conclusion (line 48-50): Based on a single proband, authors should not generalize the conclusion.

The followings are my specific comments:
  • The manuscript does not answer a specific question. It describes presence OTOG gene missense variant found in a Brazilian patient with Meniere's disease
  • This topic is relevant to Otology. It does not answer a specific gap in knowledge. 
  • It describes the presence OTOG gene missense variant found in a Brazilian patient with Meniere's disease
  • The authors should specify whether the manuscript describes a case report or a cohort study. 
  • The conclusion generalizes the finding based on one patient.
  • The references are appropriate.
  • There is no additional comments on the tables and figures.
